# Seismic Analysis Method for Underground Structure in Loess Area Based on the Modified Displacement-Based Method

**Ruijie Zhang [1], Dan Ye [1,2,\*], Jianting Zhou [1] and Dengzhou Quan [3]**

1   State Key Laboratory of Mountain Bridge and Tunnel Engineering, Chongqing Jiaotong University, Chongqing 400074, China; xjdzhangruijie@163.com (R.Z.); jtzhou@cqjtu.edu.cn (J.Z.)
2   School of Tourism Management and Services, Chongqing University of Education, Chongqing 400067, China
3   School of Civil Engineering, Chang'an University, Xi'an 710061, China; qdz0809@chd.edu.cn
\*   Correspondence: yedan@cque.edu.cn

**Abstract:** At present, the seismic design research of underground structures in loess areas is lagging behind compared with practical engineering requirements. The selection of seismic calculation methods and parameters does not consider the influences of the special geological conditions in various regions, so their usefulness is limited. Based on the above problems, a modified displacement-based method (DBM) was proposed and its application was compared with the most commonly used methods of analysis (force-based design method, displacement-based design method, detailed equivalent static analysis numerical method, and the full dynamic time-history method). The results were also validated by considering data from shaking table tests conducted on a case study involving the underground Feitian Road subway station in Xi'an. The results show that compared with DBM, the average accuracy of the modified DBM technique is improved by 41.65%. The modified DBM offers good accuracy, simplicity in its model, a rapid analysis time, and easy convergence.

**Keywords:** traffic engineering; seismic calculation; loess area; DBM; DESANM

## 1. Introduction

With the growth of population and industrial activities, the shortage of available space directly restricts the rapid development of big cities. To solve this problem, many countries have begun to increase their development and utilization of urban underground space. Earthquake damage to underground structures is thus mitigated, as the confining pressure exerted by the surrounding soil can improve the level of structural safety in the event of an earthquake. Hence, seismic calculations and seismic measures are not applied to the underground structures associated with subway systems. However, earthquake disasters in recent decades have affected thinking around this traditional concept, especially the Hanshen earthquake which damaged the 3-km long subway tunnel and five subway stations, indicating the possibility of subway underground structural damage and secondary disasters remains significant [1–3]. Subway stations are a service-oriented public facility and host a concentrated population (many service facilities), and require a long time to expedite emergency evacuation.

At present, the seismic design research of underground structures in loess areas is lagging compared with the engineering requirements, for example, the selection of design parameters related to the characteristics of loess soil lacks a clear set of rules, thus hindering the safe and efficient seismic design of underground structures in loess area.

Currently, the relevant codes for the design of underground structures include: the *Code for Seismic Design of Buildings* (GB 50011-2010) [4] and the *Code for Design of Civil Air Defense Basements* (GB 50038-2005) [5]. However, these do not provide relevant seismic calculation methods. The inertial force method for underground structures in soft soil has been given in the *Code for Seismic Calculation of Subway Building Structures* (DG/TJ08-2008) [6], but it is not necessarily suitable for those loess areas. The response acceleration

method and response displacement method provided in the *Code for Seismic Calculation of Urban Rail Transit Structures* (GB 500909-2014) [7] have no specific provisions on the horizontal relative displacement of the stratum and the foundation spring stiffness parameters. The differences of ground motion characteristics among different soil bodies are not considered in the *Code for Seismic Design of Underground Structures* (GB/T 51336-2018) and other international codes [8–17].

In recent years, many studies about seismic design and analysis of underground structures have been published, but most of them do not consider the dynamic characteristics of soil or the methods are too complex to be suitable for engineering design [18–22]. Seismic design and analysis methods mainly include coupled and decoupled approaches, numerical dynamic analyses, and quasi-static calculation methods. A soil-structure interaction analysis of underground tunnels was performed by Kisiridis in1983 [23]. The magnitude and distribution of static normal soil stresses against underground structural cylinders were studied by Penzien and Wu in 1998 [24]. New analytical solutions for a deep tunnel in a saturated poro-elastic ground were explored by Bobet A [25]. The analytical solutions for the thrust and moment in the lining of a circular tunnel due to seismic-induced ovaling deformation were studied by Park et al. [26]. An analytical solution for a rectangular opening in an infinite elastic medium subjected to far-field shear stresses was proposed for drained and undrained loading conditions [27]. The main limitations of the decoupled approaches were investigated and discussed through a large set of numerical simulations [28]. The main results of several numerical dynamic analyses of propped embedded retaining structures in the time domain were demonstrated by Soccodato FM and Tropeano G [29]. The seismic behavior of a multi-propped retaining structure was evaluated considering soil-structure interaction effects [30]. The quasi-static methods mainly include the force-based method (FBM), displacement-based method (DBM), and detailed equivalent static analysis numerical method (DESANM) [7–9,18,19,31–39]. Coupled and decoupled approaches and numerical dynamic analyses are too complex to be suitable for engineering design, and the quasi-static methods applicable to engineering design do not consider the effects of different soil properties

In conclusion, the selection of seismic calculation methods and parameters does not consider the influences of the special geological conditions in various regions, so their usefulness is limited. The seismic analysis of subway station structures in loess areas is yet to be codified. In the existing research results, the seismic performance of underground subway station structures in loess is rarely researched. For the practical engineering of subway underground structures in a loess area, there is no reference seismic experiment, reliable quasi-static calculation method, or seismic parameter-calculation method yet available.

Based on the above problems, herein, a modified DBM was proposed and its application was compared with the most commonly used methods of analysis (FDM, DBM, DESANM, and full dynamic-time history analysis (FDTHA)). The results were validated using data from a shaking table test conducted on the basis of a case study (the Feitian Road underground subway station in Xi'an). The modified DBM offers good accuracy, a simple model, rapid modeling, and easy convergence. It provides engineering designers with a seismic design and analysis method for underground structures with convenient application and high precision. Therefore, the results obtained in this study can be considered useful to designers who are required to address the seismic design of underground structures.

In Section 2, the main seismic calculation methods of underground structures were analyzed, and the methods of calculation of foundation reaction spring stiffness parameters and formation horizontal relative displacement parameters were explored. In Section 3, the shaking-table test scheme design and the FDTHA simulation modeling of a subway station in a loess area were conducted. In Section 4, the results of shaking-table testing and the FDTHA numerical simulation were studied, and the seismic calculation methods for the main underground structures were compared. In Section 5, the DBM was modified

according to the experimental results and simulation results, and the accuracy of the modified MDB was verified.

## 2. Seismic Calculation Method and Important Calculation Parameters of Subway Underground Structures

### 2.1. FBM

When the FBM is used in the seismic calculation of underground stations, the static forces such as the load exerted by the soil on the roof, the inertial force of each component, and the increment of active lateral earth pressure at each point of the external wall are often used to replace the seismic force [18,19]. The calculation model under lateral seismic action is shown in Figure 1 [31–33], where $P_i$ represents the inertial force representing the weight of the floor, $F_1$ is the inertial force of the side wall, $F_2$ denotes the sum of ground and roof overburden forces, and subscripts 1 to 3 represent the top, middle, and bottom plates, respectively; $K$ is the foundation spring stiffness, $V$ and $H$ denote the structural bottom plate and side wall, and $\Delta e$ refers to the increment of lateral earth pressure caused by the earthquake. The horizontal inertial force can be calculated according to the *Code for Seismic Calculations in Railway Engineering* (GB 50111-2006) [9].

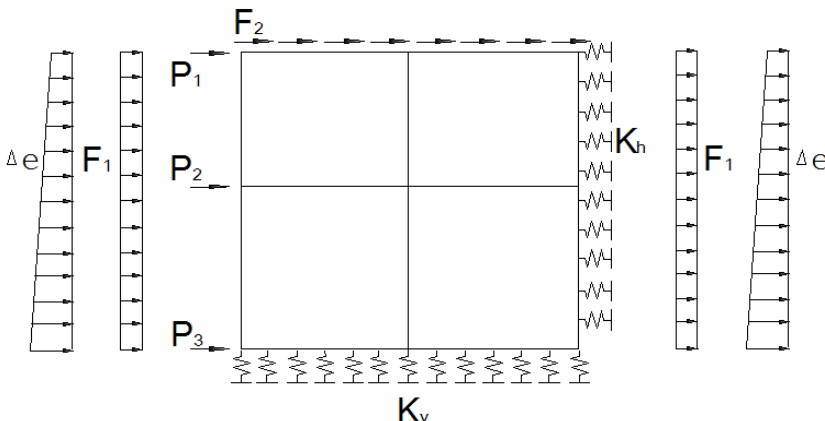

**Figure 1.** FBM calculation model.

### 2.2. DBM

The DBM [7,8] emphasizes the interaction between the soil and the underground structure. In this method, the foundation reaction spring was adopted to simulate the surrounding soil, and the horizontal relative displacement of the soil layer was applied at the end of the foundation reaction spring. The calculation model of the DBM under transverse earthquake action is illustrated in Figure 2, where 1, 2, 3, and 4 represent the ground, bedrock, soil displacement, and subway station acceleration, respectively, $K_{sv}$ is the tangential shear foundation spring stiffness of the top and bottom plates of the structure, $K_{sh}$ represents the tangential shear foundation spring stiffness of the side wall of the structure, $k_v$ is the normal compression foundation spring stiffness of the top and bottom plate of the structure, $K_h$ denotes the normal compression foundation spring stiffness of the side wall of the structure, $\tau_B$ is the friction shear force per unit area produced by the soil on the structural floor, and $\tau_U$ is the friction shear force per unit area of the soil acting on the roof of the structure.

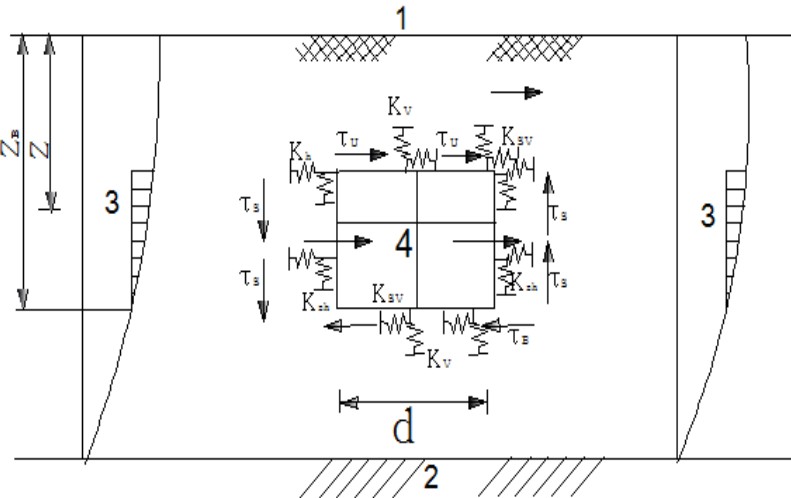

**Figure 2.** DBM calculation model.

### 2.3. DESANM

The DESANM [7,8,31,36–38] requires the engineer to establish soil and structure models at the same time in order to better reflect the interaction between soil and surrounding geotechnical media. This method reflects the phenomenon that the deformation difference between underground structure and surrounding soil changes in an irregular manner during an earthquake. The basic equation of DESANM is as shown in Equation (1).

$$[K]\{u\} = -[M](\{\ddot{u}T\} + [R]\{\ddot{u}\mathrm{g}T\}) = -[M]\{\ddot{u}T\} \tag{1}$$

The DESANM can be used in the automatic calculation of the interactive force between the soil medium and an underground structure through finite element analysis software, thus avoiding the error caused by improper selection of foundation reaction spring parameters when establishing spring stiffness values. The calculation model of DESANM under lateral earthquake action is illustrated in Figure 3, where 1 and 2 are equivalent lateral inertial accelerations of the soil and structure, respectively.

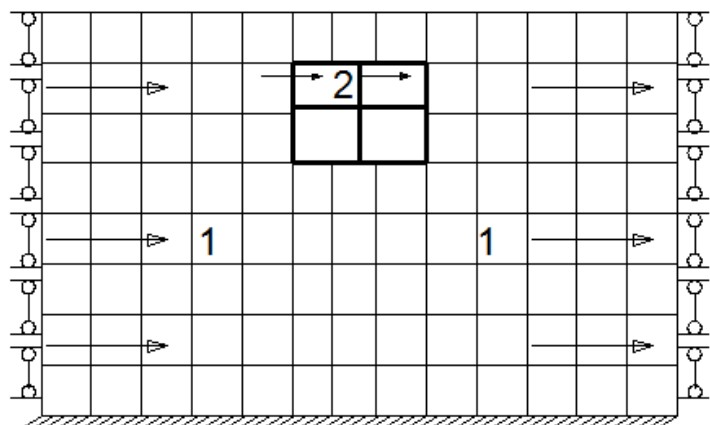

**Figure 3.** DESANM calculation model.

### 2.4. FDTHA

Although the three-dimensional model has the advantages of high calculation accuracy, the modeling is complex and the solution does not readily converge, posing a difficulty for engineering designers. The complete soil-structure system can be modeled and analyzed using 2-d numerical models. FDTHA is considered among the most sophisticated and accurate methods for the seismic analysis of underground structures [7,8,31,36–38].

The method can efficiently describe the kinematic and inertial aspects of the soil-structure interaction and the complex geometry of the soil deposit. FDTHA can be used to investigate the seismic behavior of an underground structure through numerical simulation [29,31].

### 2.5. Important Calculation Parameters

(1)    Foundation Spring Stiffness

As an important parameter of the FBM and the DBM, the foundation reaction spring stiffness parameter plays an important role in FBM and the DBM. As the key parameter of the DBM and the DESANM, the horizontal relative displacement parameter of the stratum under seismic action has a significant influence on the seismic calculation results. For the value of foundation reaction spring stiffness parameter, the relevant norms make no clear provisions, and the relevant research has reached no unified conclusion.

The finite element calculation model has been established, as shown in Figure 4. In Figure 4, the width of the soil model is six to seven times that of the underground structure, and the depth of the soil model extends to the bedrock surface. The influences of soil properties and structural shape characteristics on foundation spring stiffness in loess area were evaluated, as shown in Figure 5. The proposed equation of foundation spring stiffness was fitted using MATLAB, as shown in Equations (2)–(7). The proposed equation can be used for the seismic calculation of underground station structures with various rectangular section sizes in loess sites.

$$
\begin{aligned}
\text{K1} = &-0.1041E\left(\frac{1}{h^2} - 3.6\frac{1}{h} - 0.05\right)\left(d^3 - 17.682d^2 + 144.992d + 1459.612\right) \\
&\left(\frac{1}{b^3} - 0.15\frac{1}{b^2} - 0.003\frac{1}{b} + 0.003\right)\frac{1}{4.417-v} - 9.125 \times 10^4
\end{aligned}
\tag{2}
$$

$$
K2 = 2.696 \times 10^{-6}E\left(\frac{1}{h^2} - 9.16\frac{1}{h} - 0.18\right)\left(d^3 - 6.912d^2 - 450.422d - 9408.926\right)(0.83 + v) + 1540
\tag{3}
$$

$$
K_3 = -13.92E\frac{1}{H^{1.1}}\left(\frac{1}{b^2} - 1.794\frac{1}{b} - 0.087\right)\left(\frac{1}{1.128 - v}\right) + 8.076 \times 10^6
\tag{4}
$$

$$
K_4 = -28.15E\frac{1}{H^{0.3}}\left(\frac{1}{b^2} - 0.4\frac{1}{b} + 0.009\right)\left(\frac{1}{5.8 - v}\right) + 1.122 \times 10^7
\tag{5}
$$

$$
\begin{aligned}
K_5 = &\ 0.002413E(d^3 - 7.373d^2 - 223.914d + 136.272) \\
&\left[\left(\frac{1}{b}\right)^3 - 0.329\left(\frac{1}{b}\right)^2 + 0.007\left(\frac{1}{b}\right) - 4.9 \times 10^{-5}\right]\left(\frac{1}{0.7-v} + 6.579\right) + 805.6
\end{aligned}
\tag{6}
$$

$$
\begin{aligned}
K_6 = &\ 0.0006815E(d^3 - 18.856d^2 + 149.856d + 314.46) \\
&\left[\left(\frac{1}{b}\right)^3 - 0.299\left(\frac{1}{b}\right)^2 + 0.049\left(\frac{1}{b}\right) - 0.00026\right]\left(\frac{1}{0.7-v} + 17.158\right) - 8769
\end{aligned}
\tag{7}
$$

When $B \leq 30$ m, $H \leq 30$ m, and 3 m $\leq D \leq 3.5$ m, the following relationship holds:

$$
K_4 = (0.5 \sim 0.6)K_3
\tag{8}
$$

$$
K_2 = (0.85 \sim 0.9)K_1
\tag{9}
$$

where $K_1$ and $K_2$ represent the normal and tangential stiffness of the side, respectively (Pa); $K_3$ and $K_4$ are the normal and tangential stiffness of the bottom surface, respectively (Pa); $K_5$ and $K_6$ denote the normal and tangential stiffness of the top surface (Pa); $b$ and $h$ are the width and height of the structural section, respectively (m); $d$ refers to the burial depth of the roof (m); $H$ is the distance between the station floor and bedrock (m); and $E$ and $v$ represent the elastic modulus (Pa) and Poisson's ratio, respectively.

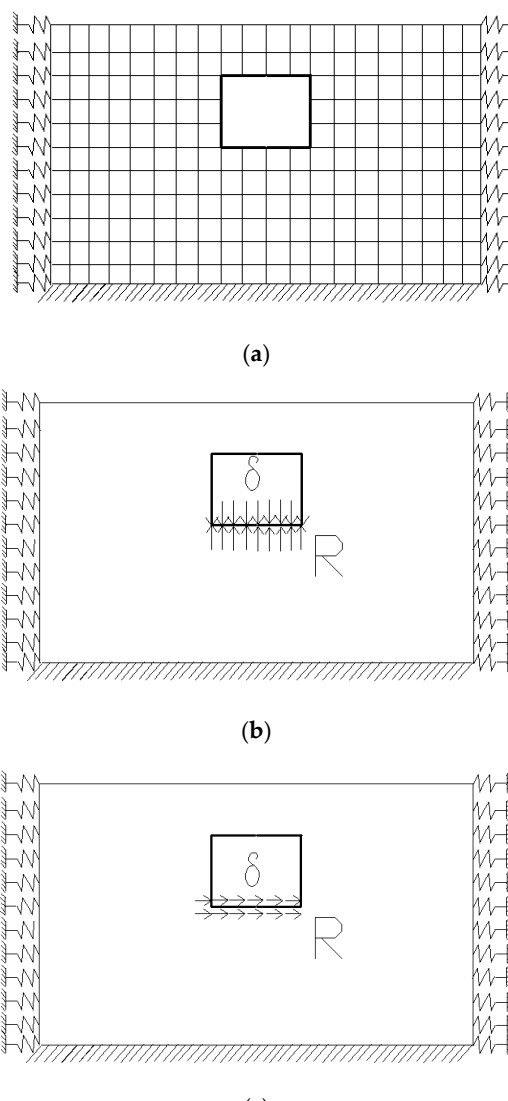

**Figure 4.** Finite element calculation model of reaction spring parameters of underground structure foundation. (**a**) Meshed model, (**b**) normal spring stiffness calculation model, (**c**) tangential spring stiffness calculation model.

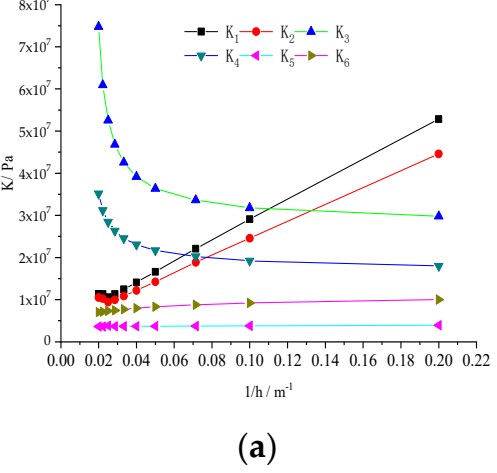

**Figure 5.** *Cont.*

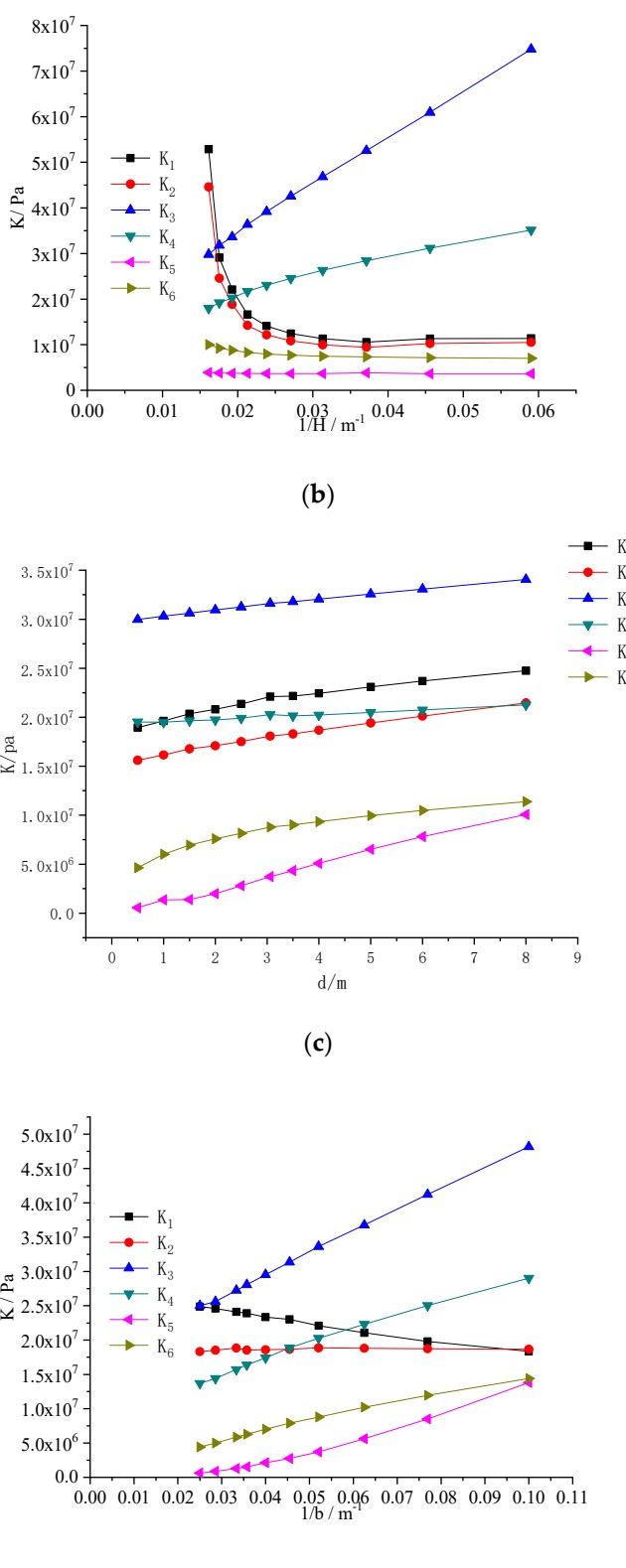

(**b**)

(**c**)

(**d**)

**Figure 5.** *Cont.*

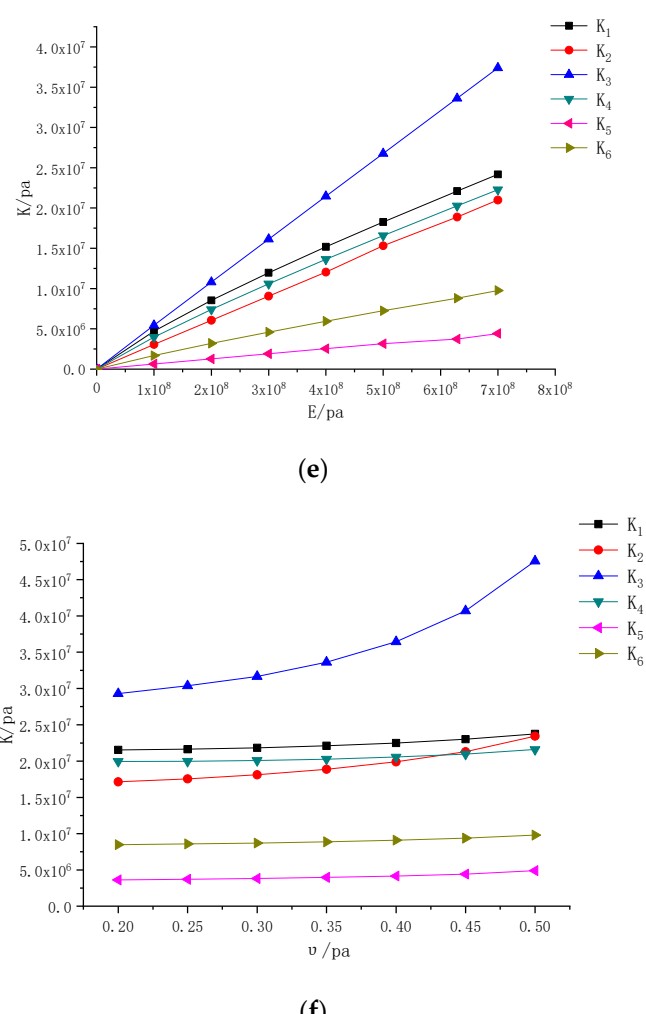

(**e**)

(**f**)

**Figure 5.** Influences of various factors on foundation spring stiffness: (**a**) *K versus* 1/*h*, (**b**) *K versus* 1/*H*, (**c**) *K versus d*, (**d**) *K versus* 1/*b*, (**e**) *K versus E*, (**f**) *K versus v*.

(2)     Horizontal Relative Displacement of Strata in a Loess Area

A numerical model of free field dynamic response considering seismic intensity, ground motion characteristics and loess soil characteristics was established, as shown in Figure 6. Where 1 is an infinite element boundary, 2 denotes a fixed boundary, and 3 is bedrock. The maximum response horizontal displacement of free field was obtained by inputting the Xi'an artificial wave, Taft wave, and Songpan wave in turn (Table 1 and Figure 7).

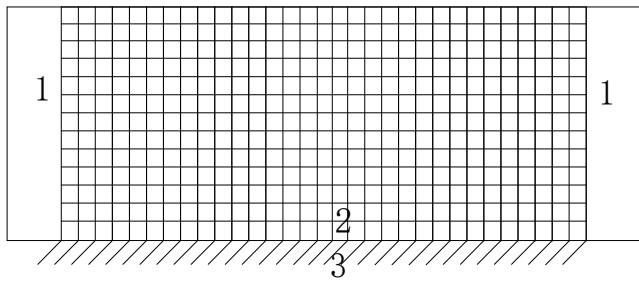

**Figure 6.** A numerical model of free field vibration.

**Table 1.** Maximum free-field horizontal displacement.

| Seismic Fortification Intensity/Degree | 7 | 7.5 | 8 | 8.5 | 9 |
|---|---|---|---|---|---|
| Peak acceleration of seismic wave/$g$ | 0.1 | 0.15 | 0.2 | 0.3 | 0.4 |
| Peak value of horizontal displacement/m | 0.251 | 0.502 | 0.528 | 0.847 | 1.141 |

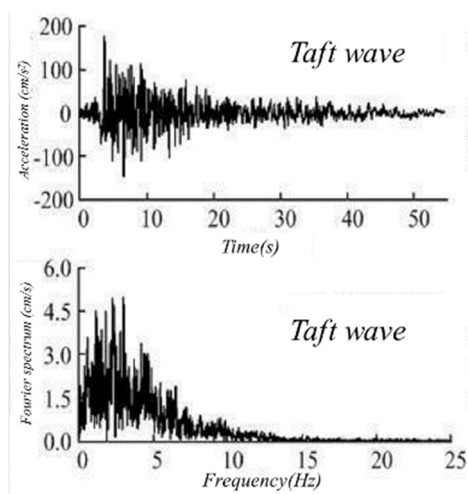

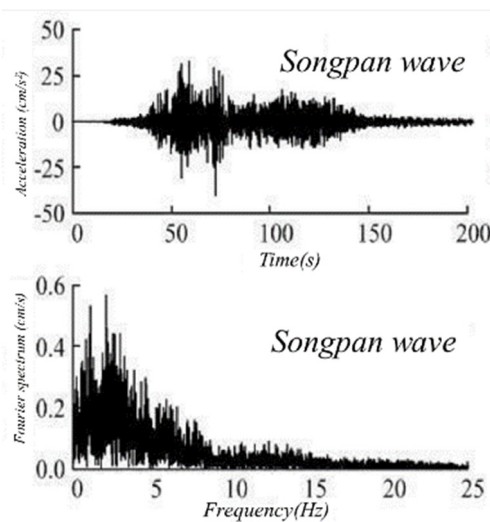

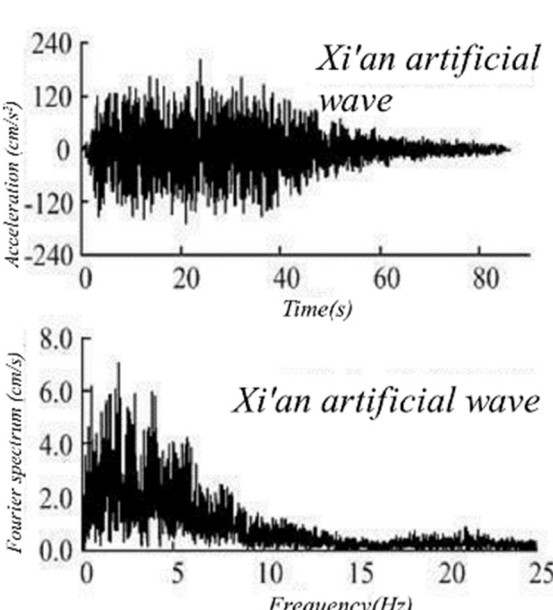

**Figure 7.** Acceleration and Fourier spectra of input ground motion.

## 3. Experiment and Simulation

### 3.1. Background to the Experiment

The geomorphic unit of Xi'an Feitian Road Station belongs to the second and third grade loess tableland. The lithologic characteristics of the site strata are summarized in Tables A1 and A2. The table is arranged from top to bottom according to the order of soil layers from shallow to deep. The underground subway station is a reinforced concrete structure with a total height of 14.01 m and a total width of 19.2 m. The longitudinal spacing of the center pillar is 9 m. The cross section of the center pillar measures 0.8 m × 1.2 m. The depth of soil above the roof is 3.459 m. The density of concrete $\rho$ is 2.5 g/cm$^3$, its modulus of elasticity $E$ is 35 GPa, and Poisson's ratio $\nu$ is 0.15. A typical cross section is shown in Figure 8.

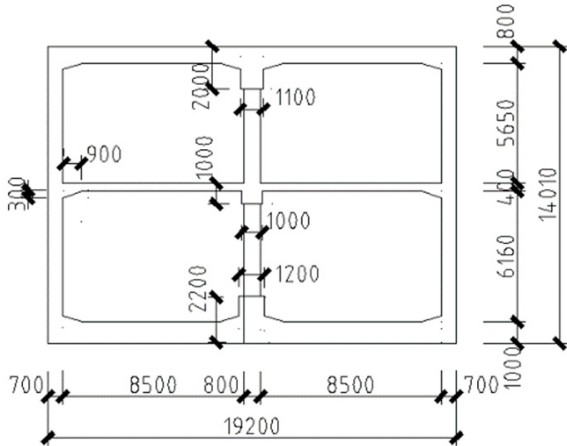

**Figure 8.** Schematic diagram of a typical cross section (dim.: mm).

The station is located in Chang'an district of Xi'an city. The site type in this area is class II, the basic seismic intensity is 8 degrees, and the characteristic period of seismic response spectrum is 0.4 s. The seismic parameters of station engineering are displayed in Table 2.

**Table 2.** Ground motion parameters.

| Position | Parameter | 50-Year Exceedance Probability | | | 100-Year Exceedance Probability | |
|---|---|---|---|---|---|---|
| | | 63% | 10% | 2% | 10% | 2% |
| Ground | Tg(s) | 0.38 | 0.43 | 0.68 | 0.59 | 0.83 |
| | Ag(g) | 0.079 | 0.235 | 0.457 | 0.340 | 0.575 |
| Floor | Tg(s) | 0.60 | 0.64 | 0.90 | 0.75 | 1.00 |
| | Ag(g) | 0.051 | 0.151 | 0.358 | 0.247 | 0.441 |

*3.2. Shaking-Table TEST Scheme*

Shaking-table tests on loess subway station models are mainly used to assess the seismic mechanism and soil structure dynamic interaction mechanism of subway station structures in loess sites. Due to the small size of the subway station model used here, it is difficult to eliminate the gravitational distortion effect by the artificial mass model with full counterweight, so an added-mass model was adopted. Based on the Buckingham π theorem, the length, elastic modulus, and acceleration were selected as basic physical quantities, and the table size, dynamic performance, bearing tonnage, and other supporting equipment performance of the test system were fully considered to ascertain the similarity relationship of the model system (Table A3). In the test, the method of sticking lead blocks into the structure was used to realize the additional artificial mass. Taking Feitian Road Station of Xi'an Metro Line 4 as the prototype structure, the subway station model was established by using particulate concrete and galvanized steel wire. The loess was taken from the foundation pit of Feitian Road Station of Xi'an Metro Line 4. When preparing the model foundation, the loess was layered into the model soil box, and the water content and density of the model foundation were controlled according to the natural water content and density of the prototype site soil. The excitation system used in this test was a horizontal one-way high-performance seismic simulation shaking table produced by MTS Company, Eden Prairie, MN, USA. The table measures 3.36 m × 4.86 m, the maximum load is 25 t, and the maximum acceleration is 1.0 g. The Taft wave, Songpan wave and Xi'an artificial wave were selected as input ground motions, respectively. Since the subway station was not situated in an active fault zone, only the transverse ground motion was input. The sensor arrangement is shown in Figure 9.

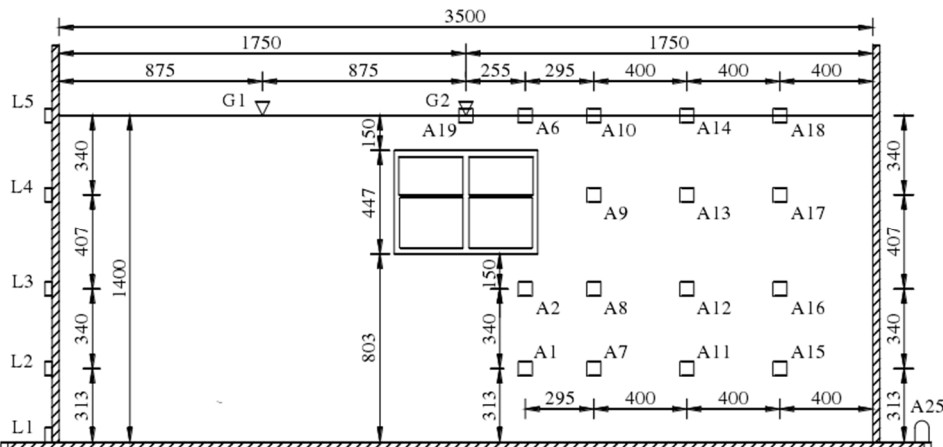

**Figure 9.** Sensor layout (dimensions: mm).

### 3.3. FDTHA Modeling

Although the three-dimensional model has the advantages of high calculation accuracy, the modeling is complex and the solution does not readily converge, posing a difficulty for engineering designers. In addition, considering the long length of the flying station model, the size and structure of each section are similar; the three-dimensional dynamic interaction system of loess subway underground structure was considered as a two-dimensional plane strain problem in the seismic analysis by a time-history analysis method.

The typical section of loess site and subway underground structure was numerically simulated, and the model was established as shown in Figure 10 [39], where 1 represents an infinite element boundary, 2 is a fixed boundary, and 3 denotes the bedrock.

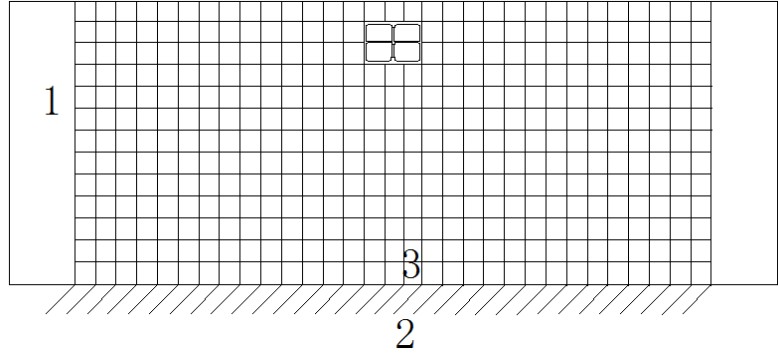

**Figure 10.** Schematic representation of the time-history analysis calculation model.

The finite element mesh was used to simulate the near-field region, and the infinite element boundary was utilized to simulate the far-field region far away from the structure. The size of the finite element mesh (Figure 10) is 150 m × 70 m (width × height), the height of the infinite element mesh shall be consistent with that of the finite element mesh, and the width is unlimited. The Taft wave, Songpan wave, and Xi'an synthetic wave were selected. The plastic damage model of concrete in ABAQUS finite element analysis software was employed to simulate the mechanical behavior of the prototype subway station concrete. The parameters used in the plastic damage model are listed in Table 3. The CPE4R (Four node bilinear plane strain quadrilateral element, reduced integration) element was used for on-site soil and the CPE4 element was used to model the subway station structure. The initial stress on the soil was calculated using the geostatic module in ABAQUS. To simulate the in-situ stress on the soil and how it affects the adjacent underground structure, the model states of excavation, support, construction of underground structure, and backfilling were established to simulate the construction process of this subway station, and the

stress state in the soil after construction was taken as the initial stress field for the later dynamic analysis.

**Table 3.** Parameters used in the plastic damage model.

| Parameter | Value | Parameter | Value |
|---|---|---|---|
| Density/kg/m$^3$ | 2500 | Angle of dilation $\psi/°$ | 30 |
| Elastic modulus $E$/MPa | $0.66 \times 10^4$ | Coefficient of viscosity $\mu$ | 0.0005 |
| Poisson's ratio $\nu$ | 0.2 | Tensile variable $\omega_t$ | 0 |
| Ultimate compressive stress/MPa | 5.39 | Compression variable $\omega_c$ | 1 |
| Invariant stress ratio $K_c$ | 0.667 | Damping ratio $\xi$ | 0.1 |

## 4. Results

### 4.1. Earthquake Damage

The seismic damage to this structure was observed by the shaking-table test (Figures 11 and 12). The column is the most severely damaged component in the subway station model. There are vertical cracks in the upper middle column, and the longitudinal reinforcement of the lower middle column is exposed with significant spalling (a typical shear-compression failure). The most severe damage to the side wall and the center column occurs at the joint with the top and bottom plates, and the seismic damage entails significant concrete spalling and the armpit angle reinforcement is pulled out. The roof and floor remain in good condition after the earthquake. The widest crack is 15 mm across and the maximum vertical differential settlement is 32 mm.

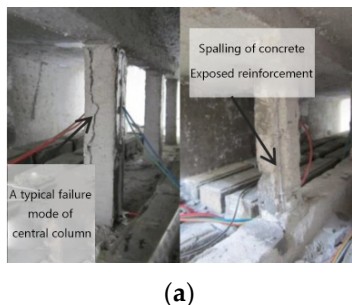 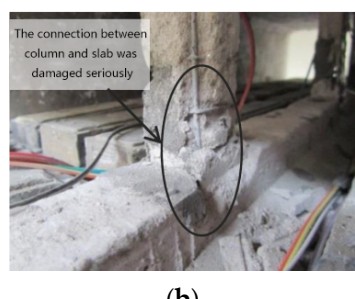 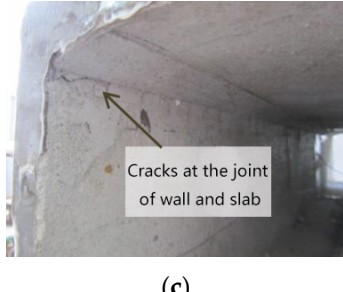

| (**a**) | (**b**) | (**c**) |
|---|---|---|

**Figure 11.** Earthquake damage phenomenon in the shaking-table test: (**a**) failure mode of the join between central column and slab, (**b**) failure mode of the joint between side wall and slab, (**c**) failure mode of the joint between side wall and slab.

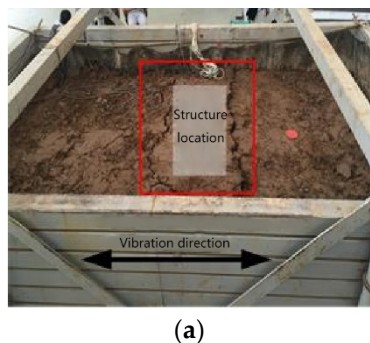 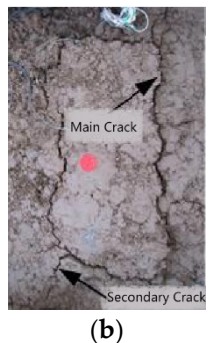 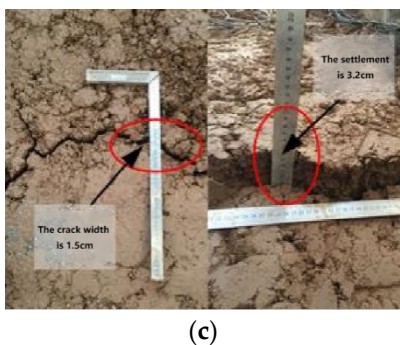

| (**a**) | (**b**) | (**c**) |
|---|---|---|

**Figure 12.** Final seismic failure phenomenon of the foundation in the model test: (**a**) macroscopic earthquake damage, (**b**) fracture distribution, (**c**) crack and settlement measurement.

### 4.2. Reliability Analysis of FDTHA

Under the action of seismic waves with different peak accelerations, when the maximum shear deformation occurs on the surface and bottom of the model foundation, the comparison of the horizontal relative displacement of each measuring point relative to the bottom of the model foundation is shown in Figure 13. The distribution of the horizontal relative displacement of the model foundation along the soil depth in the numerical simulation and shaking-table test is consistent. Acceleration sensors were, respectively, arranged along the side wall of the model structure from bottom to top. Under the action of seismic wave, the comparisons of acceleration response time history and corresponding Fourier spectrum of each measuring point in the model structure between numerical simulation and shaking-table test are demonstrated in Figure 14. The time-history waveform, amplitude, and Fourier spectrum of the acceleration response in the model structure recorded by the numerical simulation and shaking-table test are similar. Therefore, the numerical simulation and the numerical model of the dynamic interaction between the loess and a subway station established in this paper can be deemed to have simulated the acceleration response of a subway station structure reliably.

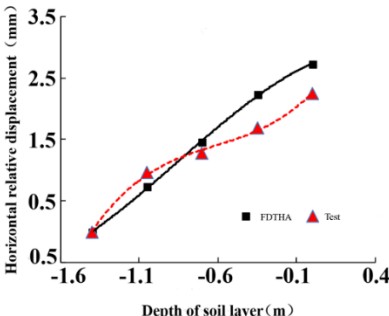

**Figure 13.** Time-history analysis and horizontal relative displacement of model foundation in the shaking-table test.

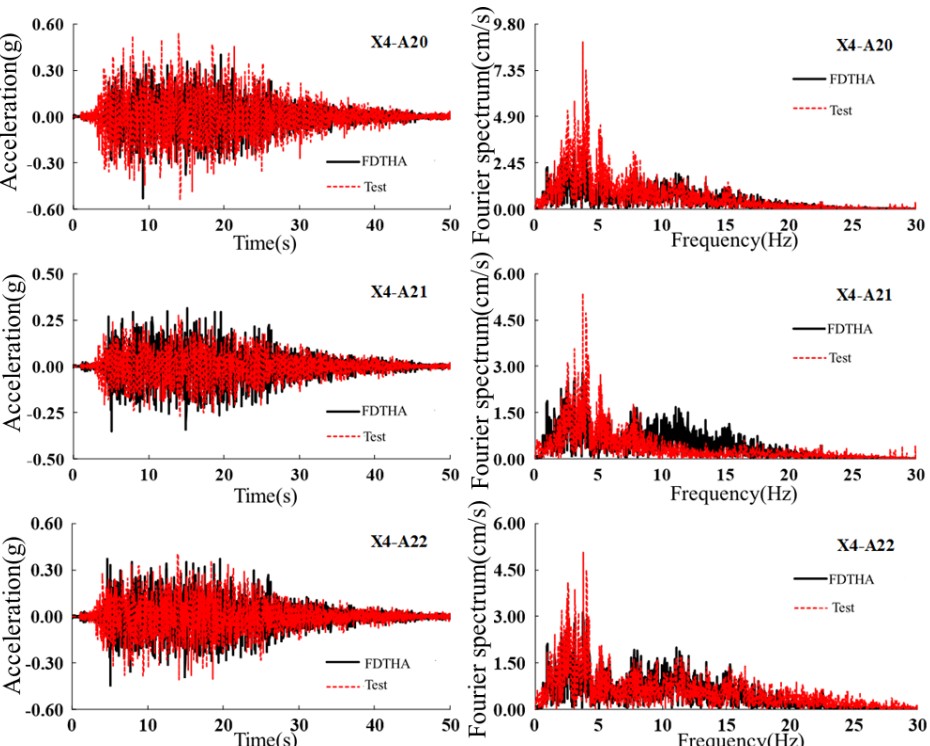

**Figure 14.** Time-history analysis and acceleration response of model structure in the shaking table test.

### 4.3. Reliability Analysis of Seismic Calculation Methods for Subway Underground Structure

The accuracy of the other three quasi-static seismic calculation methods was compared and evaluated based on the structural internal force output results of the time-history analysis method (Table A4). Observation points a and b mark the top and bottom of the upper column, respectively; c and d mark the top and bottom of the lower column, respectively; e and f mark the top and bottom of the upper wall, respectively; and g and h mark the top and bottom of the lower wall, respectively. According to the data in Table 4 and the shaking-table damage phenomenon, the difference is presented as follows:

**Table 4.** Comparison of structural internal forces in normal use.

| Observation Point | Bending Moment/kN·m | | Shear Force/kN | | Axial Force/kN | |
|---|---|---|---|---|---|---|
| | Normal Use Stage | Under Lateral Seismic Action | Normal Use Stage | Under Lateral Seismic Action | Normal Use Stage | Under Lateral Seismic Action |
| a | −0.014 | 215.6 | 0.005 | −201 | −1180 | −1031.7 |
| b | 78.2 | −153.9 | −132.000 | −284 | −1260 | −1154.5 |
| c | 0.005 | 131.6 | 0.003 | −419.6 | −1560 | −1421.9 |
| d | 90.5 | −197.5 | −134 | −517.6 | −428 | −1558.6 |
| e | −248 | −658.2 | −976 | −118.2 | −547 | −506.9 |
| f | −550 | 49.2 | 1050 | −180.9 | −727 | −794.8 |
| g | −347 | −224.9 | −183 | −272.5 | −610 | −854.3 |
| h | −418 | −1352.4 | 568 | −358.2 | −1080 | −1395 |

(1) After selecting reasonable seismic design parameters (the practical data tabulated herein were used for the horizontal relative displacement of the stratum in the loess area, and the comprehensive recommended equation proposed in this paper was used for the foundation spring stiffness), the results of the structural internal force of the DESANM are more consistent with the time-history analysis method. The FBM that neglects the seismic shear force leads to underestimation of the internal forces. The bending moment and shear force output by DBM are closest to those predicted by use of the time-history analysis method, but the accuracy of the axial force output remains insufficient.

(2) From the shaking-table test, the top of the upper column is the most severely damaged, and the bottom of the lower column follows. The results of time-history analysis and DESANM are consistent with the damage seen in the shaking-table test; the maximum bending moment on the middle column is located at the top of the upper column, followed by the bottom of the lower column. Therefore, the comprehensive recommended equation and practical tabulated data better reflect the effect of soil on the structure.

### 4.4. Seismic Analysis

(1) The internal forces on an underground structure, as calculated by FBM, are small, which is related to the fact that the friction and shear stresses caused by soil deformation are not considered in the FBM. The DBM is simple to establish, and the principle is to simulate the interaction between soil and structure by establishing springs and applying horizontal displacement to the key stratum.

(2) The DBM simulates the interaction between soil and structure by establishing springs and applying horizontal displacement to a stratum, and the model is simple. The correct values of spring stiffness and ground horizontal displacement make the output of structural shear force and bending moment approach those calculated using the time-history method of analysis. The output accuracy of shear force and bending moment values is higher than that of DESANM, but the accuracy of axial force output is poorer than that of DESANM.

(3) The loess stratum is established in the calculation model of the DESANM, so the calculation relies on fewer parameters. When the value of horizontal relative displacement is accurate, the results of structural internal force of DESANM are consistent with that of the time-history method of analysis. However, the calculation is more onerous, ranking second only in complexity compared to that in the time-history analysis.

## 5. Modification of DBM

To provide the designers of such an underground subway station with good seismic calculation accuracy and a simple calculation model, the modified DBM was used to make the axial force output more accurate.

### 5.1. Influences of Transverse Seismic Force on Subway Underground Structure

Only dead load and live load are input into the time-history analysis model to obtain the structural internal forces acting on the station in its normal service conditions, therefore, the influences of transverse seismic forces thereon can be determined.

It can be seen from Table 4 that, differing from the trend in shear force results and bending moment output, the axial forces are similar to those predicted using FDTHA. Therefore, the axial forces on such an underground structure are greatly affected by vertical load and less affected by transverse seismic load. According to the earthquake damage phenomenon affecting such a system in loess deposits, the soil still undergoes vertical displacement under the action of transverse earthquake excitation, and then generates vertical seismic earth pressures on the underground structure. Therefore, the neglect of the vertical seismic earth pressure of soil on the structure is the reason why the axial force predicted by DBM is too small. For this reason, the vertical seismic earth pressure of soil on the structure and the correct calculation of vertical earth pressure were incorporated in the original DBM calculation model.

### 5.2. Vertical Seismic EARTH Pressure

The tendency of the soil mass to undergo vertical relative displacement with horizontal displacement was revealed through shaking-table testing and analysis of the numerical model of the dynamic response to a transverse seismic wave. The results of the vertical relative displacement are listed in Table 5. The origin is set at the surface, and the change of vertical relative displacement along the burial depth ($z$) is assumed to be a cosine function $\mu_{a1(z)}$, as shown in Equation (10).

$$u_{a1}(z) = \frac{1}{2} u_{\text{max2}} \bullet \cos \frac{\pi z}{2H_b}$$ (10)

where the $\mu_{\text{max2}}$ is the peak vertical relative displacement of the stratum, $H_b$ is the burial depth to the bedrock surface (dimension: m).

**Table 5.** Peak vertical displacements of the soil mass.

| Seismic Fortification Intensity/Degree | 7 | 7.5 | 8 | 8.5 | 9 |
|---|---|---|---|---|---|
| Peak acceleration of seismic wave/g | 0.1 | 0.15 | 0.2 | 0.3 | 0.4 |
| Peak value of surface displacement/mm | 10.5 | 13.3 | 16.2 | 19.7 | 24.1 |

The earth pressure generated by the soil on the top plate of the structure under earthquake was calculated by using Equation (11).

$$N_1 = K \bullet \mu_2(z)$$ (11)

where $K$ is the stiffness parameter of the normal foundation reaction spring of the structural roof (Pa), $N_1$ denotes the vertical seismic earth pressure at the roof ($N$), and $z$ is the burial depth of the roof (m).

*5.3. Earth Pressure Acting on the Floor*

The base plate exerts pressure on the foundation, and its reaction force is the support force of the foundation on the base plate. The support force is in line with the base pressure in the opposite direction. The supporting force is called the bottom plate pressure, and the calculation method is consistent with the base pressure, as shown in Equation (12).

$$N_2 = \gamma d \tag{12}$$

where $N_2$ represents the bottom plate pressure (N), $d$ denotes the burial depth of the bottom plate (m), $\gamma$ is the bulk unit weight of the soil mass (N/m$^3$).

*5.4. Reliability Analysis: Modified DBM*

The comparison of structural axial force values after correction by DBM is embodied by Table 6. The accuracy of the axial force predicted by the modified DBM is greatly improved, and the discrepancy is within 16%, and the average error is 6.65%, which agrees with the axial force results of the FDTHA method. The error in DBM axial force result is as high as 86.47%, and the average error is 48.30%, which is related to the problem whereby the DBM technique ignores the soil to estimate the vertical seismic earth pressure and vertical earth pressure on the structure. Compared with DBM, the accuracy of our modified DBM is improved by 80.17%, and the average error is reduced by 41.65%. The modified DBM overcomes the problem whereby the DBM ignores the soil to estimate the vertical seismic earth pressure and vertical earth pressure of the structure, which leads to the underestimation of the structural axial force. This method improves the calculation accuracy of the DBM and provides a simple and fast calculation method with good seismic calculation accuracy for designers of underground subway stations.

**Table 6.** Comparison of axial force predicted by the modified DBM.

| Observation Point | | Modified DBM/kN | DBM/kN | FDTHA/kN | Error in Modified DBM | Error in DBM |
|---|---|---|---|---|---|---|
| Central column | a | −1172.70 | −510.95 | −1031.70 | 13.70% | −50.47% |
| | b | −1184.20 | −522.45 | −1154.50 | 2.60% | −54.75% |
| | c | −1399.20 | −712.10 | −1421.90 | −1.60% | −49.92% |
| | d | −1411.90 | −724.80 | −1558.60 | −9.40% | −53.50% |
| Flank | e | −509.40 | −388.26 | −506.90 | 0.50% | −23.41% |
| | f | −671.30 | −739.00 | −794.80 | −15.50% | −7.02% |
| | g | −823.80 | −330.18 | −854.30 | −3.60% | −61.35% |
| | h | −1307.50 | −188.70 | −1395.00 | −6.30% | −86.47% |

## 6. Conclusions

(1) A new analysis method useful for the evaluation of the seismic behavior of underground structures in loess area was proposed based on the DBM and it is named the modified DBM. The DBM was modified according to the results obtained from a shaking-table test and numerical simulations. The results show that the modified displacement-based method improves the accuracy of structural axial force output and compensates for the defects of the DBM. The modified DBM is applicable to any typology of underground structure and it can be applied to the seismic design of underground structure.

(2) The modified DBM was compared with the most commonly used methods of analysis (FBM, DESANM, DBM, and FDTHM). The results were validated considering using data from a shaking table test based on the Feitian Road underground subway station in Xi'an. The comparison of output results shows that the modified DBM offers good accuracy, a simple model, rapid modeling, and easy convergence.

(3) Based on the finite element analysis method, a practical table of horizontal relative displacements of strata in loess areas, and the comprehensive recommended equation

of foundation reaction spring stiffness, were provided. The practical table and the comprehensive recommended equation provide a method of estimation of those seismic calculation parameters required by designers of underground subway stations in loess areas.

(4) Due to the special properties of loess, the modified DBM is suitable for the seismic response analysis of underground structures in loess area. However, whether it is applicable to other soils remains to be studied in future research.

**Author Contributions:** Conceptualization, R.Z. and D.Q.; methodology, D.Y.; Project administration, J.Z. All authors have read and agreed to the published version of the manuscript.

**Funding:** This study was supported by Chongqing Postdoctoral Special Fund (020919014) and The Natural Science Foundation of Chongqing, China (cstc2020jcyj-bshX0118) and National Natural Science Foundation of China (U20A20314).

**Institutional Review Board Statement:** Not applicable.

**Informed Consent Statement:** Not applicable.

**Data Availability Statement:** All data are available from the author.

**Conflicts of Interest:** The authors declare no conflict of interest.

## Appendix A

**Table A1.** Physico-mechanical properties of soil layer.

| Number | Soil | Weight Density/kN/m³ | Elastic Modulus/MPa | Cohesion/kPa | Internal Friction Angle/° | Liquidity Index |
|---|---|---|---|---|---|---|
| 3-1-1 | New loess | 16.2 | 7 | 27 | 24.5 | 0.17 |
| 3-2-1 | Paleosol | 17.4 | 10 | 45 | 24 | 0.15 |
| 4-1-1-1 | Old loess | 16.1 | 9 | 35 | 23 | 0.3 |
| 4-2-1 | Paleosol 2 | 17.6 | 11 | 44 | 23 | 0.31 |
| 4-1-1-2 | Old loess 2 | 16.9 | 10 | 36 | 22.5 | 0.44 |

**Table A2.** Soil properties with increasing depth.

| Number | Soil | Thickness/m | Equivalent Shear Wave Velocity/m/s | Density/kg/m³ |
|---|---|---|---|---|
| 3-1-1 | New loess | 7.3 | 223.8 | 1620 |
| 3-2-1 | Paleosol | 3 | 335.5 | 1740 |
| 3-1-1 | New loess | 8.8 | 348.4 | 1620 |
| 3-2-1 | Paleosol | 4.2 | 354 | 1740 |
| 3-1-1 | New loess | 3.3 | 360 | 1620 |
| 3-2-1 | Paleosol | 3.6 | 361.7 | 1740 |
| 3-1-1 | New loess | 3.6 | 371.9 | 1620 |
| 4-2-1 | Paleosol 2 | 5.1 | 384.9 | 1760 |
| 3-1-1 | New loess | 2.6 | 403.8 | 1620 |
| 4-2-1 | Paleosol 2 | 2.3 | 422.8 | 1760 |
| 4-1-1-1 | Old loess | 4 | 422.8 | 1610 |
| 4-2-1 | Paleosol 2 | 2.1 | 458.5 | 1760 |
| 4-1-1-1 | Old loess | 3.2 | 465 | 1610 |
| 4-2-1 | Paleosol 2 | 2.5 | 465.3 | 1760 |
| 4-1-1-1 | Old loess | 8 | 475.7 | 1610 |
| 4-2-1 | Paleosol 2 | 1.6 | 483.5 | 1760 |
| 4-1-1-2 | Old loess 2 | 4.8 | 488.6 | 1690 |

**Table A3.** Similar constants in the test model.

| Physical Property | Physical Quantity | Similarity Relationship | Similarity Constant |
|---|---|---|---|
| Geometric characteristics | Length $l$ | $S_l$ | 1/30 |
| | Area $A$ | $S_A = S_l^2$ | 1/900 |
| | Linear displacement $l$ | $S_l$ | 1/30 |
| Material characteristics | Elastic modulus $E$ | $S_E$ | 1/5 |
| | Stress $\sigma$ | $S_\sigma = S_E$ | 1/5 |
| | Density $\rho$ | $S_\rho = S_E/(S_l S_\alpha)$ | 3.0 |
| | Quality $m$ | $S_m = S_\sigma S_l^2/S_\alpha$ | $1.11 \times 10^{-4}$ |
| Load performance | Point force $F$ | $S_F = S_\sigma S_l^2$ | $2.22 \times 10^{-4}$ |
| | Line load $q$ | $S_q = S_\sigma S_l$ | $6.67 \times 10^{-3}$ |
| | Moment $M$ | $S_M = S_\sigma S_l^3$ | $7.41 \times 10^{-6}$ |
| Dynamic characteristics | Time $t$ | $S_t = S_l^{0.5} S_\alpha^{-0.5}$ | 0.13 |
| | Velocity $v$ | $S_t = S_l^{0.5} S_\alpha^{0.5}$ | 0.26 |
| | Acceleration $\alpha$ | $S_\alpha$ | 2.0 |

**Table A4.** Comparison of internal forces at control points of structures with different calculation methods.

| Observation Point | | Bending Moment/N·m | | | | Shear Force/N | | | | Axial Force/N | | | |
|---|---|---|---|---|---|---|---|---|---|---|---|---|---|
| | | FDTHA | FBM | DBM | FBM | FDTHA | FBM | DBM | FBM | FDTHA | FBM | DBM | FBM |
| Central column | a | 215,555 | 10,423.3 | 238,803 | 279,008 | −49,184 | −4141.44 | −44,832 | −69,461.7 | −1,031,720 | −4141.44 | −510,953 | −1,045,410 |
| | b | −153,906 | −6832.71 | −181,235 | −198,130 | −67,739 | −4141.44 | −44,832 | −178,655 | −1,154,460 | −4141.44 | −522,453 | −1,140,410 |
| | c | 131,605 | −2701.63 | 121,486 | 363,861 | −43,857 | 741.297 | −109,800 | −92,933.5 | −1,421,870 | 741.297 | −712,100 | −1,297,840 |
| | d | −197,523 | 930.728 | −214,180 | −615,633 | −50,276 | 741.297 | −109,800 | −261,956 | −1,558,590 | 741.297 | −724,804 | −1,408,240 |
| Flank | e | −658,209 | −66,093.9 | −677,207 | −435,546 | 769,365 | −99,877.4 | 659,369 | −532,570 | −506,949 | −99,877.4 | −388,257 | −536,758 |
| | f | 49,262 | 45,317.8 | 43,402 | 502,996 | −143,782 | 29,138.5 | −724,622 | −178,655 | −794,779 | 29,138.5 | −738,997 | −772,260 |
| | g | −224,909 | −30,203.7 | −251,916 | −18,654.4 | 231,968 | −40,416.1 | 661,587 | −248,115 | −854,289 | −40,416.1 | −330,183 | −855,921 |
| | h | −1,352,410 | −147,243 | −1,768,540 | 363,861 | −1,995,360 | 100,578 | −1,435,260 | −2,190,510 | −1,394,980 | 100,578 | −1,887,020 | −1,411,240 |

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
