# Peer review of "Seismic Analysis Method for Underground Structure in Loess Area Based on the Modified Displacement-Based Method"

_applsci, doi:10.3390/app112311245_

Round 1

Reviewer 1 Report

English can be improved.

Literature review can be improved.

Author Response

Thanks, the English of this manuscript initially was revised by a native English-speaker engaged through the auspices of a professional proofreading service.  

The literature review has been improved and a paragraph has been added to Section 1. For details, please refer to paragraph 4 of section 1.

Reviewer 2 Report

The paper "Seismic Analysis Method for Underground Structure in Loess Area Based on the Modified Displacement-Based Method" presents a new analysis method useful for the evaluation of the seismic behaviour of underground structures in loess area. The proposed method is based on the Displacement-Based Method and its application is compared with the most used analysis method (force-based design method, displacement-based design method, detailed equivalent static analysis numerical method and full dynamic time-history method). The results are validated considering also a shaking table test carried out on a case study: the underground subway station "Feitian Road" in Xi'an. The approach is clearly described in the manuscript showing the original aspects of the research. For these reasons it is opinion of this reviewer that the paper can be considered  for the publication on Applied Sciences after the following improvements:

  • Introduction (Section 1): (i) consider as reported in 10.1016/S0886-7798(01)00051-7 for the seismic design of underground structures
  • Section 2.4: consider as reported in 10.1016/j.engstruct.2020.110497 and 10.32075/17ECSMGE-2019-0015 for the application of FDTHA considering 2D numerical model. (ii) Furthermore, in the sentence "The complete soil-structure system can be modelled and analyzed using 2D numerical models", it is important to clarify the limits and advantages compared to implementing a 3D model
  • Section 2.5: (i) Eq.4, H1.1 is correct? - (ii) improve the quality of the Figure 4 and indicate the dimensions of the model - (iii)  improve the quality of the Figure 5 - (iv) insert a figure which represents the inputs 
  • Section 3.2: in Figure 8 indicate the u.m. of the dimensions 
  • Section 3.3: (i) in Figure 9 indicate the dimensions of the model - (ii) indicate the parameters used in the plastic damage model 
  • Section 5.2: standardize the terms used in Eq. 10 with those indicated in the text
  • Conclusion (Section 6): better highlight the original aspects of the work
  • Improve the quality of the English 

Author Response

  • Introduction (Section 1): (i) consider as reported in 10.1016/S0886-7798(01)00051-7 for the seismic design of underground structures
  • Thanks.The literature review has been improved and a paragraph has been added to Section 1. For details, please refer to paragraph 4 of section 1.
  • Section 2.4: consider as reported in 10.1016/j.engstruct.2020.110497 and 10.32075/17ECSMGE-2019-0015 for the application of FDTHA considering 2D numerical model.
  • References 29 and 31 have been cited in Section 2.4.
  • (ii) Furthermore, in the sentence "The complete soil-structure system can be modelled and analyzed using 2D numerical models", it is important to clarify the limits and advantages compared to implementing a 3D model
  • Please refer to sections 2.4 and 3.3 for details.
  • Section 2.5: (i) Eq.4, H1.1 is correct? 
  • Yes,error of H1.1 is smaller compare to H1.0  .
  • (ii) improve the quality of the Figure 4 and indicate the dimensions of the model -
  •  The quality of the Figure 4 has been improved,and the second paragraph of Section 2.5 has been added with regard to the dimensions of the model.
  • (iii)  improve the quality of the Figure 5 
  •  The quality of the Figure 5 has been improved.
  • (iv) insert a figure which represents the inputs
  • The problem has been modified,please See Figure 7 for details.
  • Section 3.2: in Figure 8 indicate the u.m. of the dimensions 
  • The problem has been modified,please See Figure 9 for details.
  • Section 3.3: (i) in Figure 9 indicate the dimensions of the model
  • See the third line of the third paragraph of Section 3.3 for details.
  •  indicate the parameters used in the plastic damage model
  • See the tabel 3 for details.
  • Section 5.2: standardize the terms used in Eq. 10 with those indicated in the text
  • Please refer to the second paragraph of Section 5.2 for details.
  • Conclusion (Section 6): better highlight the original aspects of the work
  • The problem has been solved, see section 6
  • Improve the quality of the English 
  • Thanks, the English of this manuscript initially was revised by a native English-speaker engaged through the auspices of a professional proofreading service.    

Reviewer 3 Report

The manuscript investigates and compares different seismic analysis methods (force-based method, displacement-based method, detailed equivalent static analysis method) for underground structures. In particular, the displacement-based method (DBM) was modified according to the results obtained from shaking-table tests and numerical simulations.

The topic addressed in the paper is worthy of investigation, the work is well-organized, the methodology employed is adequately presented.

It is the opinion of this reviewer that the manuscript can be considered for publication in Applied Sciences journal.

Some suggestions are given to the authors:

1) Abstract should be improved, better indicating the reasons of the research problem and the main novelty aspects of the work.

2) In Introduction, the main objectives and the main novelty aspects of the study should be better presented and discussed. Moreover, it is suggested to better discuss the main contribution into practice.

3) In Introduction, considering the topic addressed in the manuscript, it is suggested to mention the following reference, which investigates seismic analysis methods of underground structures:

https://doi.org/10.1016/j.engstruct.2020.110497

4) It is required to specify if the results obtained and the displacement-based method (DBM) proposed in this study can be extended to other typologies of underground structures.

5) A more comprehensive discussion about the accuracy of the results obtained using the modified displacement-based method (DBM) should be inserted.

6) In Conclusion (Section 6), no recommendations have been given regarding future studies. Such recommendations constitute an important element for the development of the research topic.

7) The manuscript is characterized by a low quality of the English language. An extensive review of the whole text should be carried out.

Author Response

1) The problem has been corrected. Please see the Abstract section

2) The problem has been corrected. Please see the Introduction section

3)  The following reference has been cited.Please see the Introduction section.

4) The problem has been corrected.Please see the  section 6(1).

5) The problem has been corrected. Please see the section 5.4.

6) The problem has been corrected.Please see the  section 6(4).

7) Thanks, the English of this manuscript initially was revised by a native English-speaker engaged through the auspices of a professional proofreading service; in this revision,  the manuscript has been carefully checked and revised again, highlighted in Track Change in Microsoft Word.We now believe that the paper should meet the standards required for publication in your journal and should make our work accessible to the scientific community.
